# Construction of Oxidative Stress-Related Genes Risk Model Predicts the Prognosis of Uterine Corpus Endometrial Cancer Patients

**DOI:** 10.3390/cancers14225572

**Published:** 2022-11-14

**Authors:** Qin Liu, Minghua Yu, Tao Zhang

**Affiliations:** 1Department of Pathology, Women’s Hospital School of Medicine Zhejiang University, Hangzhou 310000, China; 2Department of Gynaecology, Women’s Hospital School of Medicine Zhejiang University, Hangzhou 310000, China

**Keywords:** uterine corpus endometrial carcinoma, oxidative stress, risk model, prognosis, TCGA

## Abstract

**Simple Summary:**

Uterine corpus endometrial carcinoma (UCEC) is the fifth most common malignancy and has become one of the most frequent gynecological cancers in women. In UCEC, a typical symptom of irregular vaginal bleeding usually occurs, leading to the massive release of heme. Then, oxidative stress can be mediated by the degradation products of heme and thus accelerate the development and occurrence of tumors. There are few pieces of research associated with oxidative stress in UCEC. This study aimed to explore the potential link between oxidative stress and UCEC. We discovered 136 oxidative stress-related differentially expressed genes (DEGs) in UCEC, from which we screened 25 prognostic genes significantly related to the overall survival of UCEC patients. Then, a 7-OSRGs-based risk score (H3C1, CDKN2A, STK26, TRPM2, E2F1, CHAC1, MSX1) was generated by Lasso regression. In summary, our results demonstrated that the signature based on OSRGs could serve as a reliable biomarker for predicting the clinical outcome in UCEC.

**Abstract:**

Oxidative stress contributes significantly to cancer development. Recent studies have demonstrated that oxidative stress could alter the epigenome and, in particular, DNA methylation. This study aimed to explore the potential link between oxidative stress and uterine corpus endometrial carcinoma (UCEC). An analysis of RNA-seq data and relevant clinical information was conducted with data from The Cancer Genome Atlas (TCGA), and oxidative stress genes were obtained from Gene Set Enrichment Analysis (GSEA). Differentially expressed genes (DEGs) in normal and tumor groups of UCEC were analyzed using GO and KEGG enrichment analysis. As a result of survival analysis, Lasso regression analysis of DEGs, a risk score model of oxidative stress-related genes (OSRGs) was constructed. Moreover, this study demonstrated that OSRGs are associated with immune cell infiltration in UCEC, suggesting oxidative stress may play a role in UCEC development by activating immune cells. We discovered 136 oxidative stress-related DEGs in UCEC, from which we screened 25 prognostic genes significantly related to the overall survival of UCEC patients. BCL2A1, CASP6, GPX2, HIC1, IL19, MSX1, RNF183, SFN, TRPM2 and HIST1H3C are associated with a good prognosis while CDKN2A, CHAC1, E2F1, GSDME, HMGA1, ITGA7, MCM4, MYBL2, PPIF, S100A1, S100A9, STK26 and TRIB3 are involved in a poor prognosis in UCEC. A 7-OSRGs-based risk score (H3C1, CDKN2A, STK26, TRPM2, E2F1, CHAC1, MSX1) was generated by Lasso regression. Further, an association was found between H3C1, CDKN2A, STK26, TRPM2, E2F1, CHAC1 and MSX1 expression levels and the immune infiltrating cells, including CD8 T cells, NK cells, and mast cells in UCEC. NFYA and RFX5 were speculated as common transcription factors of CDKN2A, TRPM2, E2F1, CHAC1, and MSX1 in UCEC.

## 1. Introduction

Uterine corpus endometrial cancer (UCEC) is the most common gynecological malignancy of the female reproductive tract worldwide, especially in advanced countries [1]. Histologically, UCEC can be classified into three subtypes such as endometrioid, serous, or clear cell, of which endometrioid endometrial carcinoma is the major histological type [2,3]. Although women with UCEC usually have a relatively good prognosis, those with high-risk disease characteristics are more likely to experience recurrences [4]. Due to limited treatment options and the poor prognosis of advanced endometrial cancers, an understanding of the genetic drivers of treatment vulnerability, prognostic predictors, and resistance is crucial [5].

Oxidative stress is due to an imbalance between the production of reactive oxygen species and the antioxidant capacity [6]. Recent research has reported oxidative stress is involved in multiple pathological conditions, such as degenerative diseases, aging, and cancer [7,8]. Signaling pathways associated with cell proliferation are demonstrated to be affected by oxidative stress [9]. As an example, the signaling pathways for epidermal growth factor receptors and its pivotal proteins, including RAS/RAF, MEK, protein kinase C, the mitogen-activated protein kinases ERK1/2, nuclear factor erythroid 2-related factor, phospholipase C and phosphatidylinositol 3-kinase are affected by oxidative stress [10,11]. Moreover, ROS also altered the p53 suppressor gene, which functions as a hub factor in apoptosis. In summary, oxidative stress alters gene expression, cell proliferation, and apoptosis [12,13,14].

It has long been believed that the generation of excessive ROS in response to oxidative stress can lead to cell death and tissue damage, potentially causing cancer [15]. Nevertheless, studies on the effects of oxidative stress on cancer have yielded conflicting results. In breast cancer, it is likely that miR-526b and miR-655 contribute to the induction of oxidative stress by regulating TXNRD1 expression [16], the overexpression of which promotes the invasive capacity of breast cancer cells. In prostate cancer, IL-8 activates the mTOR signaling pathway to defend prostate cancer cells against the oxidative damage induced by GSK-3β [8]. In lung cancer, TWIST2 induces oxidative stress in cancer cells by regulating the FGF21-mediated AMPK/mTOR signaling pathway and prevents lung cancer from progressing [17]. In endometrial cancer, Punnonen et al. discovered superoxide dismutase activity was significantly lower in cancer tissues than in normal endometrium, suggesting that endometrial cancer was related to a damaged enzymic antioxidant defense system [18]. Further, Marta et al. demonstrated a modulated response to oxidative stress facilitated the invasion of endometrial cancer via ETV5 [19]. Besides, oxidative stress has been proven to play an important role in the pathogenesis and progression of endometriosis [20]. Levels of oxidation-related markers are markedly elevated in patients with endometriosis [21]. Subsequently, the upregulation of antioxidant functions in endometriosis may lead to the restoration of cell survival and follow-up malignant transformation [22]. Up to now, studies on the prognostic significance of oxidative stress-related genes in UCEC are very limited. Thus, examining how oxidative stress impacts tumor prognosis by investigating the OSRGs in the UCEC could provide useful therapeutic guidance.

In this work, we obtained RNA sequencing (RNA-seq) data from the TCGA dataset and ultimately identified 136 differentially expressed OSRGs and Gene Ontology (GO), and Kyoto Encyclopedia of Genes and Genomes (KEGG) were then used to analyze the mechanism of action of OSRGs in UCEC. Then, we constructed a prognostic model based on 7-OSRGs (H3C1, CDKN2A, STK26, TRPM2, E2F1, CHAC1, MSX1) by Lasso regression. A comparison of the mutation landscapes was made between high-risk and low-risk groups. Moreover, we conducted an immune-infiltration analysis and transcription factor prediction. In summary, our results demonstrated that the signature based on OSRGs could serve as a reliable biomarker for predicting the clinical outcome in UCEC.

## 2. Materials and Methods

### 2.1. Data Acquisition

From The Cancer Genome Atlas (TCGA) website, we downloaded RNA-Seq data and clinicopathological information of 587 UCEC samples, containing 35 normal samples and 552 tumor samples (https://portal.gdc.cancer.gov/projects/TCGA-UCEC, accessed on 10 July 2022). The OSRGs (n = 666) were obtained from the online GSEA website (http://www.gsea-msigdb.org/gsea/index.jsp, accessed on 10 July 2022) [23]. A total of 548 patients with UCEC were included in the analysis after downloading the corresponding clinical and survival information from the TCGA cohort. R software was used to process and normalize the raw reads.

### 2.2. Differential Gene Expression Screening

By using the “limma” dataset software package, we identified the differentially expressed genes (DEGs) in the UCEC samples and normal samples in the TCGA cohort. Then, we also identified the expression of OSRGs in UCEC samples by the “limma” package. Criteria for screening were a log_2_|FC| of >1.5 and adjusted *p* values < 0.05, which were defined as the oxidative stress-related DEGs. A Venn diagram was used to illustrate the results, and the R packages “heatmap” and “volcanoes” were utilized to visualize the DEGs.

### 2.3. Functional Enrichment Analysis of DEGs

Analysis of multiple genes was conducted to identify possible biological processes and enrichment pathways of DEGs by using GO and KEGG. Go database focuses on gene characteristics involving cell components, biological processes, and molecular functions at different dimensions and levels. KEGG is a kind of gene annotation database that integrates genomics, chemistry, and systemic functional information. Statistics were considered significant when the adjusted *p*-value was less than 0.05.

### 2.4. The Prognostic Values of the Oxidative Stress-Related DEGs

In order to determine the key genes associated with the prognosis of patients with UCEC, The TCGA database was searched for the acquisition of prognostic and clinicopathological data of 545 patients with UCEC. The R package “survminer” and “survival” were used to screen prognostic-related genes in UCEC. Further, Kaplan-Meier survival analysis was used to evaluate whether OSRGs influenced overall survival (OS) among patients with UCEC. A *p*-value < 0.05 was used as the screening standard.

### 2.5. Development of a Prognostic Gene Signature Based on the Oxidative Stress-Related DEGs

Lasso-Cox regression is one of the most commonly used methods for high-dimensional predictor selection. In the current study. A Lasso Cox regression model was analyzed using the R package “glmnet.” Based on the minimum criteria, seven genes with nonzero coefficients were selected, and the risk score of a gene signature was calculated using the formula below: Risk Score = $\sum_{i}^{n}X_{i}\times Y_{i}$ (*X*: coefficient of each gene, *Y*: gene expression level). The median risk score of each patient was used to determine which patients were low-risk and high-risk. Next, we compared the OS time and survival possibility between groups at low and high risk by the Kaplan-Meier analysis. A 2, 3, 5 year ROC curve was constructed to evaluate the risk score’s sensitivity and specificity by the “survival,” “survminer,” and “timeROC” R packages.

### 2.6. Somatic Mutation Analysis between Two Subgroups

UCEC somatic mutation data were downloaded from the TCGA Data Portal in “maf” format. The “Maftools” package in R software was used to generate waterfall plots. This tool enabled the visualization and summarization of the mutation between low-risk and high-risk groups.

### 2.7. Immune Landscape Differences between Two Subgroups

In order to analyze immune characteristics in UCEC, the expression data were loaded into CIBERSORT (https://cibersort.stanford.edu/, accessed on 15 July 2022) and repeated 1000 times to calculate the proportion of 22 different types of immune cells [12]. Ultimately, our results are presented in a landscape map that shows the relative proportion of 22 immune cell types between the low- and high-risk groups.

### 2.8. Differential Analysis of Immune Cell Infiltration and Immune Checkpoint between Two Subgroups

A CIBERSORT approach combined with the LM22 signature matrix was used to determine differences in immune infiltration between the 2 subtypes. Based on the ESTIMATE algorithm, the stromal scores, immune scores and the ESTIMATE score of TCGA-UCEC samples were calculated. Genes related to immune checkpoints were acquired according to previous research [24]. In addition, we utilized the “GSVA” package to explore the correlation between 7-OSRGs expression and 22 kinds of immune infiltrating cells.

### 2.9. Verification of Prognosis-Related 7-OSRGs Expression

To verify the protein expression of OSRGs expression in normal and tumor tissues, we analyzed data from Human Protein Atlas (HPA) (https://www.proteinatlas.org/, accessed on 15 July 2022) and determined whether the differences in protein expression levels were consistent with the previous mRNA expression from TCGA.

### 2.10. Prediction of Transcription Factors of 7 OSRGs in UCEC

To investigate the upstream regulatory mechanism of the 7 OSRGs selected based on Lasso regression, an online tool, CHEA3 (https://maayanlab.cloud/chea3/, accessed on 1 August 2022), was used to predict the transcription factors of 7 OSRGs in UCEC.

## 3. Results

### 3.1. Identification of Oxidative Stress-Related DEGs in UCEC

As a whole, 7905 DEGs were identified with the criteria of log_2_|FC| > 1.5 and a *p*-value < 0.05, whereas the expression pattern of DEGs in normal samples and tumor samples is visualized with volcano maps (Figure 1B). The Venn diagram (Figure 1C) showed that 136 differentially expressed OSRGs were screened from the overlap of OSRGs and DEGs, and the expression pattern of differentially expressed OSRGs is shown in the hierarchical heatmap (Figure 1A).

### 3.2. Functional Analysis of Differentially Expressed OSRGs

The most distinct GO category with significant enrichment of OSRGs was identified by GO enrichment analysis. GO annotation revealed that the differentially expressed OSRGs were involved in the cellular intrinsic apoptotic signaling pathway, DNA packaging, chromatin assembly or disassembly, cell adhesion molecule binding, cadherin binding, etc. (Figure 2A–C). Subsequently, we performed KEGG analysis and revealed that the OSR DEGs contributed to systemic lupus erythematosus, viral carcinogenesis transcriptional misregulation in cancer, etc. (Figure 2D and Table 1).

### 3.3. Identification of OSR DEGs Related to the Prognosis in UCEC

To identify the hub genes associated with prognosis in patients with UCEC, we screened 4647 prognostic-related genes in UCEC, which were overlapped with 136 differentially expressed OSRGs by the Venn diagram (Figure 3A). Eventually, 25 differentially expressed OSRGs related to the prognosis in UCEC were screened from the overlap of OSRGs and DEGs. Among these, the results showed that 10 OSR DEGs, including BCL2A1, CASP6, GPX2, HIC1, IL19, MSX1, RNF183, SFN, TRPM2 and HIST1H3C, are associated with a good prognosis while 13 OSR DEGs including CDKN2A, CHAC1, E2F1, GSDME, HMGA1, ITGA7, MCM4, MYBL2, PPIF, S100A1, S100A9, STK26 and TRIB3 are involved in a poor prognosis in UCEC (Figure 3B).

### 3.4. Construction and Prognostic Value of Differentially Expressed OSRGs

A total of 25 differentially expressed OSRGs were related to prognosis. An analysis of Lasso regression was performed in order to prevent the prognostic model from being overfit (Figure 4A–C). Eventually, the corresponding seven OSRGs were selected for the model, which were H3C1, CDKN2A, STK26, TRPM2, E2F1, CHAC1, and MSX1. Furthermore, we were able to calculate the corresponding regression coefficients, which were −0.2438, 0.1117, 0.1156, −0.2271, 0.1264, 0.0195, and −0.0834, respectively. By using the formula above in combination with the beta value of multivariate Cox regression, the risk score formula is as follows: Risk-score = (−0.2438) × H3C1 + (0.1117) × CDKN2A + (0.1156) × STK26 + (−0.2271) × TRPM2 + (0.1264) × E2F1 + (0.0195) × CHAC1 + (−0.0834) × MSX1.

According to the above formula, each UCEC patient’s risk score was directly determined. Next, patients were categorized into the high-and low-risk subgroups. High-risk patients’ survival rates were lower than those of low-risk patients, according to a Kaplan-Meier analysis. (Figure 5A, log-rank *p* < 0.001; HR = 3.83, 95% CI = 2.55−5.75). The area under the curve (AUC) for predicting 2-, 3-, and 5-year OS was 0.661, 0.689, and 0.689, respectively (Figure 5B). As a result, the seven-gene risk model had high accuracy for predicting the OS of UCEC patients.

### 3.5. Mutation Landscape Associated with OSRGs Risk Scores 

We compared the mutation landscapes for high- and low-risk groups. Results showed that TP53 mutation events were more frequent in samples with higher risk scores, while more mutation events of PTEN/PIK3/ARIDIA occurred in samples with lower risk scores. (Figure 6).

### 3.6. Immune Status Analysis

An analysis of the association between OSRGs-related risk and immune status was conducted in the TCGA cohort. The low-risk group exhibited considerably elevated percentages of CD8 T cells. There was a stronger tumor immune response in the low-risk group compared to the high-risk group (Figure 7B). Immune, ESTIMATE, and stromal scores were used to assess the tumor immune microenvironment, illustrating that OSRG-related risks and immune reactions are negatively correlated (Figure 7C). In addition, compared to the low-risk group, the high-risk group had significantly lower expression of immune checkpoint-related genes such as CTLA4, HAVCR2, TIGIT, and PDCD1 (Figure 7D). 

In addition, a spearman correlation analysis showed a correlation between H3C1 expression and Th2 cells, NK cells, Th17 cells, and aDC (Figure 8). CDKN2A expression level correlated with macrophages, CD8 T cells, aDC, iDC, NK cd56dim cells, DC, pDC, T cells, T helper cells, Tcm, Th17 cells, and Th2 cells. STK26 expression level correlated with the Cytotoxic cells, Macrophages, neutrophils, NK CD56bright cells, NK CD56dim cells, NK cells, pDC, T cells, T helper cells, Tcm, Tem, Tgd, Th17 cells, Th2 cells, and TReg. TRPM2 expression level correlated with the aDC, B cells, Cytotoxic cells, DC, iDC, Neutrophils, NK CD56bright cells, NK CD56dim cells, pDC, T cells, Tcm, Th1 cells, Th17 cells, Th2 cells and TReg. E2F1 expression level correlated with the aDC, CD8 T cells, Cytotoxic cells, Eosinophils, iDC, Mast cells, Neutrophils, NK CD56bright cells, NK cells, pDC, T cells, TFH, Th17 cells and Th2 cells. CHAC1 expression level correlated with the aDC, Eosinophils, Macrophages, Mast cells, T helper cells, and Tcm. MSX1 expression level correlated with aDC, B cells, DC, Macrophages, NK CD56bright cells, NK cells, Th1 cells, Th17 cells and Th2 cells (Figure 8).

### 3.7. Immunohistochemistry Verification of 7 OSRGs Expression 

Based on the HPA data, compared with normal tissues, the expression of OSRGs, including H3C1, CDKN2A, E2F1, CHAC1, and MSX1, was significantly upregulated, except STK26 and TRPM2 (Figure 9A,B).

### 3.8. Identification of Transcription Factors of 7 OSRGs in UCEC

To investigate the upstream regulatory mechanism of the 7 OSRGs selected based on Lasso regression, the CHEA3 website (https://maayanlab.cloud/chea3/, accessed on 10 August 2022) was used to identify transcription factors involved in regulating their expression. In total, 4 transcription factors (CTCF, FOS, NFYA, RFX5) were found, and all showed significant correlations with the 7 OSRGs (Figure 10A–C). Among the four transcription factors enriched, NFYA and RFX5 showed significantly higher correlation than others, while analysis of expression profiles indicated NFYA and RFX5 were significantly upregulated in UCEC compared to normal samples (Figure 10D, *p* < 0.001). Moreover, high expression of NFYA indicated a worse survival outcome of UCEC (Figure 10E).

## 4. Discussion

Endometrial cancer (UCEC) is the fifth most common malignancy and has become one of the most frequent gynecological cancers in women [25]. The Cancer Genome Atlas (TCGA) project proposed four groups of endometrial cancer ranging from good to poor prognoses: (1) POLE-mutated (POLEmut); (2) hyper-mutated “MSI”; (3) low copy number “NSMP;” and (4) a high number of copies “TP53 mutated” (serous-like) [26].In addition to the excellent prognostic value of the TCGA molecular classification shown for UCEC, deep myometrial invasion (DMI) has been considered a crucial risk factor and influenced the risk of recurrence independently from the TCGA groups [27]. To further improve the clinical applicability of the TCGA classification, A combination of MSH6 and PMS2 may allow reducing the cost without a decrease in diagnostic accuracy, Raffone et al. concluded the combination of MSH6 and PMS2 presented the same sensitivity and specificity as four MMR proteins (MLH1, MSH6, PMS2 and MSH2) [28]. Furthermore, Santoro et al. integrated the TCGA molecular classification in the ESGO/ESTRO/ESP guidelines of endometrial carcinoma and offered new insights for UCEC treatment [29]. Apart from the molecular impact on prognosis, Troisi et al. conducted a metabolomic approach for endometrial cancer screening, which made it possible for the early and non-invasive diagnosis of UCEC by detecting circulating biomarkers [30]. Despite the potential link between oxidative stress and UCEC, there have been few studies focusing on oxidative stress in UCEC up to now.

Oxidative stress (OS) occurs when oxidative and antioxidant processes are out of balance. Various cell types are capable of accumulating persistent ROS, such as endometrial epithelial cells, stromal cells, oocytes and vascular endothelial cells. Internal signaling pathways damage cells and tissues either directly or indirectly, which causes a variety of issues with the female reproductive system [31]. Studies have revealed that oxidative stress and polycystic ovary syndrome (PCOS) are closely connected [32]. Further studies reported that women with PCOS have significantly lower ER, which is closely associated with hyperandrogenemia, metabolic disturbances, and intestinal flora imbalance, ultimately leading to an endometrial oxidative stress imbalance [33]. In UCEC, a typical symptom of irregular vaginal bleeding usually occurs, leading to the massive release of heme. Then, oxidative stress can be mediated by the degradation products of heme and thus accelerate the development and occurrence of tumors [34,35]. However, there are relatively few studies available on endometrial oxidative stress.

In this study, we first identified differentially expressed OSRGs in UCEC based on the TCGA-UCEC project. By performing survival, ROC, and Cox analyses, we found that H3C1, CDKN2A, STK26, TRPM2, E2F1, CHAC1, and MSX1 significantly correlated with the prognosis of patients with UCEC, and thus the seven-gene prognostic model was constructed. There is a significant difference in prognosis between the high- and low-risk groups.

As mentioned above, the seven OSRGs were obtained from the online GSEA website (http://www.gsea-msigdb.org/gsea/index.jsp, accessed on 15 August 2022). As reported in the available literature, CDKN2A, STK26, TRPM2, E2F1 and CHAC1 were associated with oxidative stress. For instance, CDKN2A deficiency inhibits oxidative stress, and the hypermethylation of CDKN2A is correlated with oxidative stress [36,37]. STK26 can be activated by oxidative stress and relocated to the cell periphery and subsequently promotes ezrin/radixin/moesin phosphorylation [38]. TRPM2 channel is proven to play a key role in oxidative stress-induced changes in intracellular Ca^2+^ and Zn^2+^ homeostasis to mediate oxidative stress-induced cell death [39,40]. E2F1-deficient cells are hypersensitive to oxidative stress due to a defect in cell cycle arrest, and E2F1 sumoylation serves as a protective cellular mechanism during oxidative stress response [41,42]. CHAC1 is tightly associated with oxidative stress and apoptosis by degrading glutathione [43].

Based on the seven-genes risk score generated by Lasso regression, patients were categorized into the high-and low-risk subgroups. The mutation landscapes between the two groups were compared. We found TP53 mutation events were more frequent in samples with higher risk scores, while more mutation events of PTEN/PIK3/ARIDIA occurred in samples with lower risk scores, tentatively demonstrating that our risk model is closely related to the progression of UCEC.

Recently, immunotherapy has become increasingly important for cancer treatment. Physically, the endometrial immune system differs from other immune systems for its two sides: while it protects from infections and sexually transmitted pathogens, it also allows for the implantation of allogenic embryos [44]. In the current study, the immune reaction was negatively correlated with the OSRGs-related risk. Specifically, the expression of CTLA4, HAVCR2, TIGIT, and PDCD1 was significantly lower in the high-risk group. In addition, the low-risk group exhibited considerably elevated percentages of CD8 T cells. There is a solid rationale for using CTLA-4 inhibitors in UCEC, but clinical data about two anti-CTLA-4 monoclonal antibodies, like Ipilimumab and Tremelimumab, have not been reported so far. Moreover, we also demonstrated that the expression levels of the seven OSRGs (H3C1, CDKN2A, STK26, TRPM2, E2F1, CHAC1, and MSX1) significantly correlated with Th2 cells, NK cells, Th17 cells, and aDC levels.

In addition, NFYA and RFX5 were speculated to be common transcription factors of 7 OSRGs in UCEC by the CHEA3 website (https://maayanlab.cloud/chea3, accessed on 15 August 2022). Pan et al. determined that LINC01016–miR-302a-3p/miR-3130-3p/NFYA/SATB1 axis played an essential role in endometrial cancer occurrence [45]. Eugenia et al. recently showed that NF-YA is overexpressed in liver hepatocellular, lung and head and neck squamous cell carcinomas [46,47,48]. In the current study, NFYA showed potential as a therapeutic target for UCEC treatment through general regulation of the expression of CDKN2A, TRPM2, E2F1, CHAC1 and MSX1, which requires further experimental validation and clinical trials.

## 5. Conclusions

This study utilized bioinformatics analysis to explore the roles of the OSRGs in UCEC progression. In conclusion, our study constructed a seven-OSRGs model generated by Lasso and confirmed its good prediction accuracy. We demonstrated that oxidative stress-related genes have the potential to be new biomarkers in predicting the prognosis of UCEC. There are several strengths of this study, including the large sample size, the lengthy follow-up period, and the extensive prognostic information in the TCGA database. The limitation of this study is the lack of quantitative analysis. Although H3C1, CDKN2A, E2F1, CHAC1 and MSX1 expressions in HPA data were coincident with the TCGA database, further experimental validation and clinical trials should be performed, and an extensive amount of tissue and prognosis data is required for the verification of the risk score model.

## Figures and Tables

**Figure 1 cancers-14-05572-f001:**
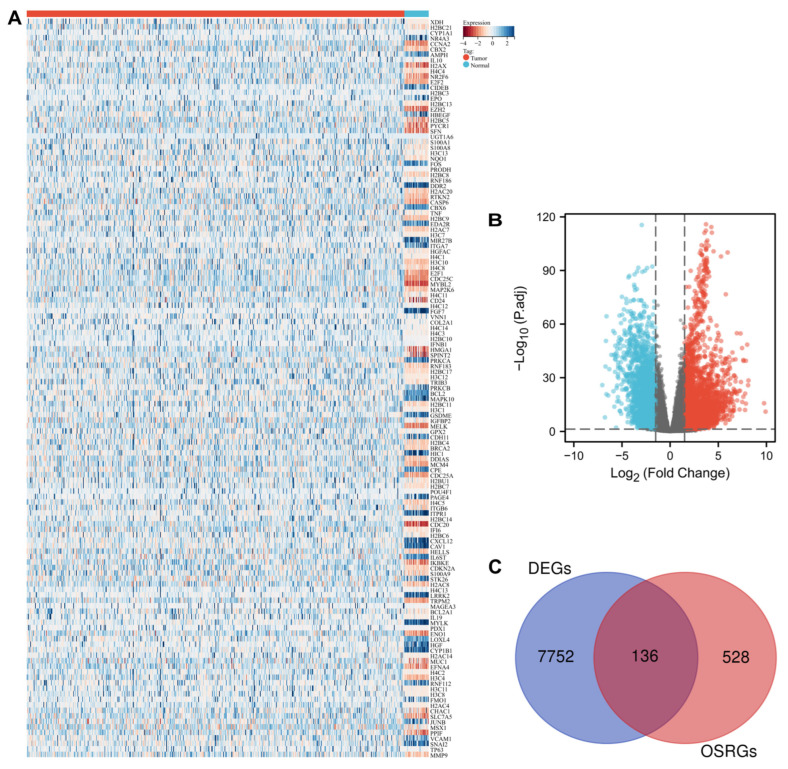
The DEGs associated with oxidative stress visualized with statistical significance: (**A**) heatmap of 136 differentially expressed oxidative stress-related genes in patients with UCEC; (**B**) Volcano plot of 7905 differentially expressed genes. (**C**) Venn diagram of the intersection of DEGs and oxidative stress-related genes. Note: DEGs: differentially expressed genes.

**Figure 2 cancers-14-05572-f002:**
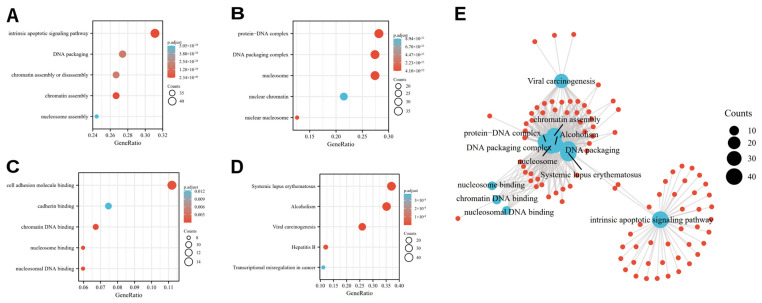
Functions and mechanisms of oxidative stress-related DEGs using GO and KEGG analysis: (**A**) biological process; (**B**) cell composition; (**C**) molecular function; (**D**) signaling pathways. (**E**) interaction between functions and mechanisms. Note: DEGs: differentially expressed genes; BP: biological process; CC: cell composition; MF: molecular function; GO: Gene Ontology; KEGG: Kyoto Encyclopedia of Genes and Genomes.

**Figure 3 cancers-14-05572-f003:**
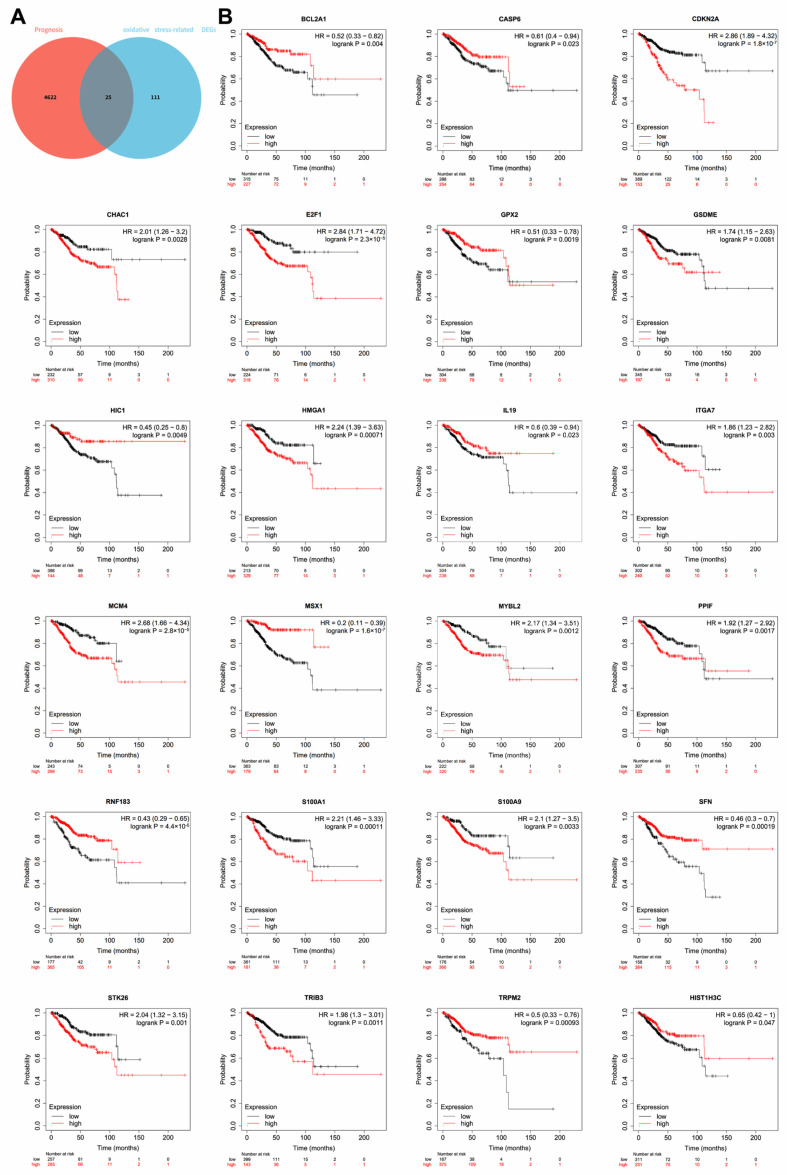
(**A**) Venn diagram of the intersection of prognostic genes in UCEC and differentially expressed OSRGs; (**B**) 23 oxidative stress-related DEGs assess the overall survival of UCEC. Note: UCEC: uterine corpus endometrial carcinoma; OSRGs: oxidative stress-related genes.

**Figure 4 cancers-14-05572-f004:**
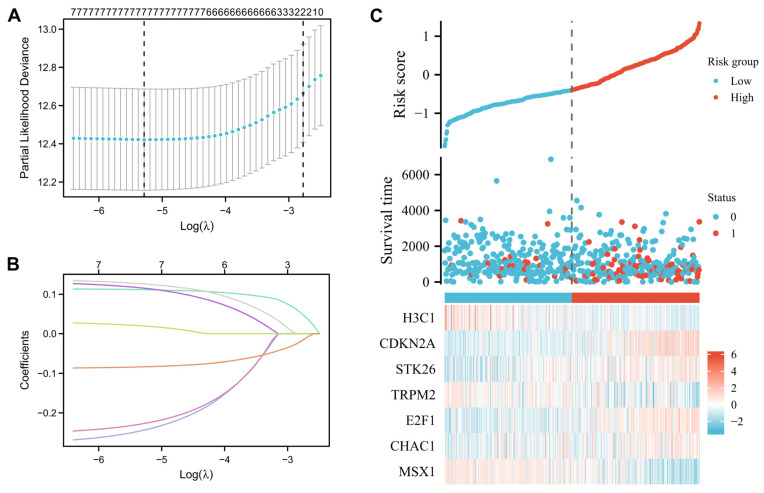
Construction of risk model based on the 7 oxidative stress-related DEGs: (**A**) Ten-time cross-validation for tuning parameter selection in the Lasso model; (**B**) Lasso coefficient profiles; (**C**) The risk score, survival status, and heat map of 7 oxidative stress-related DEGs in patients with UCEC. Note: UCEC: uterine corpus endometrial carcinoma; DEGs: differentially expressed genes.

**Figure 5 cancers-14-05572-f005:**
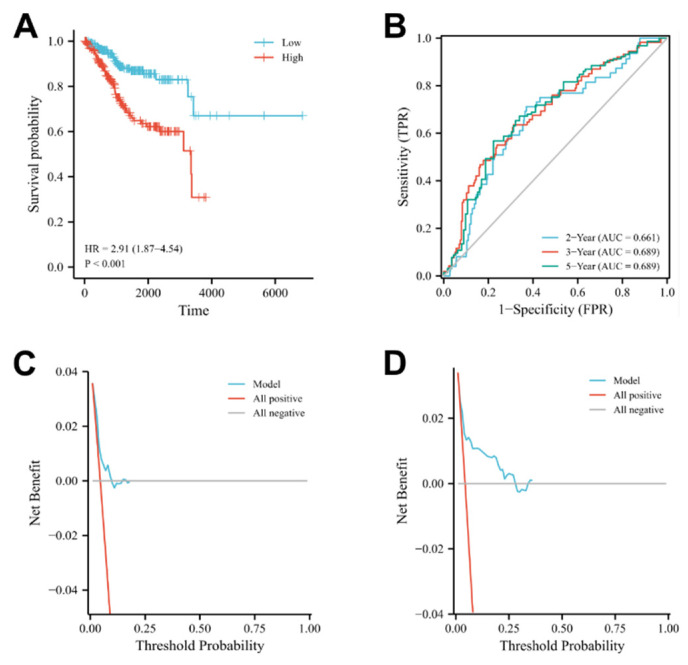
Evaluation of the 7 oxidative stress-related DEGs: (**A**) Kaplan-Meier curves show that OS was significantly different between the high- and low-risk groups in TCGA-UCEC; (**B**) The signature is shown by the time-dependent ROC curve for predicting 2, 3, and 5-year survival; (**C**,**D**) Decision curve analysis for the evaluation of the net benefits of seven oxidative stress-related DEGs model at1 years. Note: UCEC: uterine corpus endometrial carcinoma; DEGs: differentially expressed genes.

**Figure 6 cancers-14-05572-f006:**
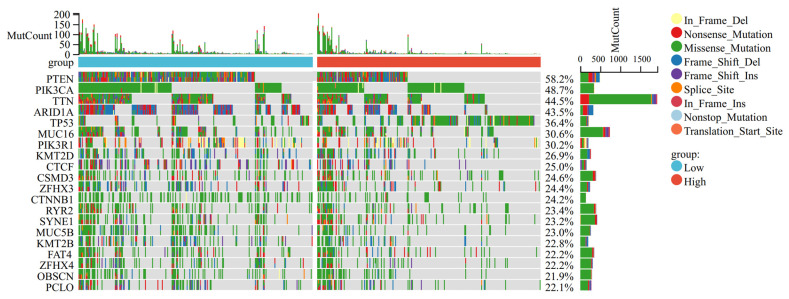
Comparison of somatic mutations between different risk subtypes.

**Figure 7 cancers-14-05572-f007:**
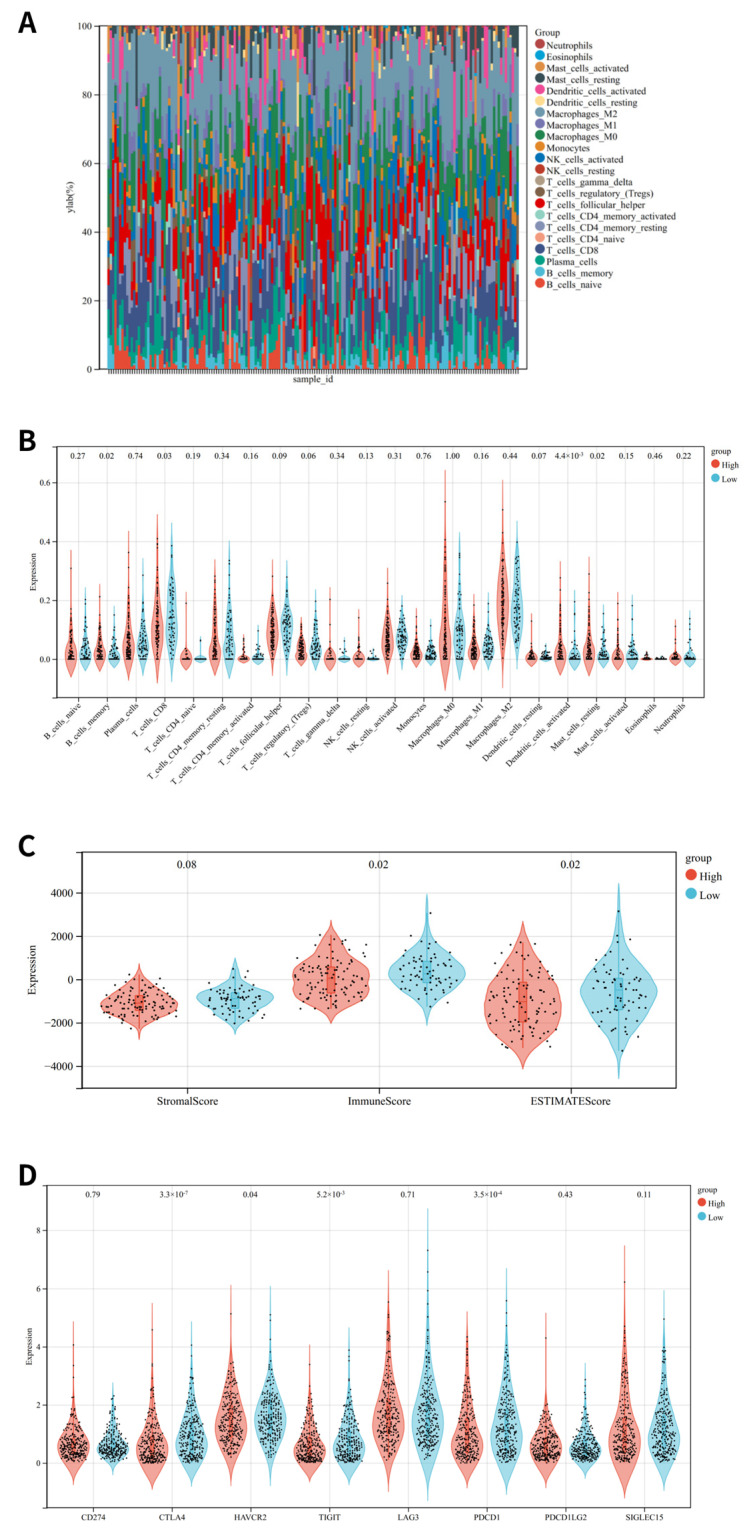
The immune landscape of risk-high and low subtypes: (**A**) Relative proportion of immune infiltration in TCGA samples; (**B**) Violin plot visualizes significantly different immune cells between different subtypes; (**C**) Violin plots show the stromal score, immune score and ESTIMATE score; (**D**) Violin plot present differential expression of multiple immune checkpoints.

**Figure 8 cancers-14-05572-f008:**
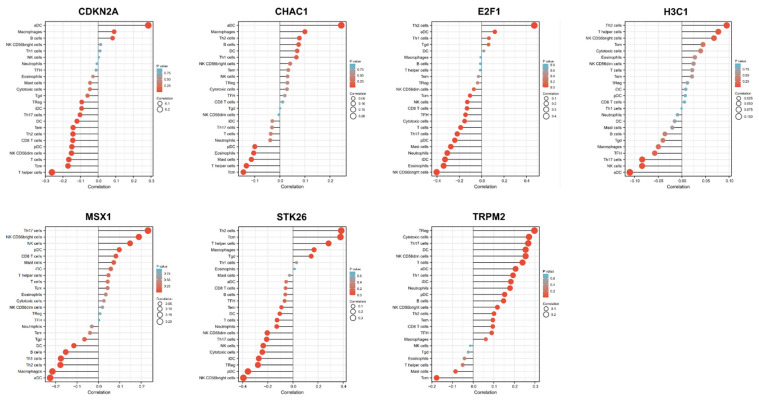
The correlation between the expression level of H3C1, CDKN2A, E2F1, CHAC1, STK26, TRPM2 and MSX1 with 22 immune cell types.

**Figure 9 cancers-14-05572-f009:**
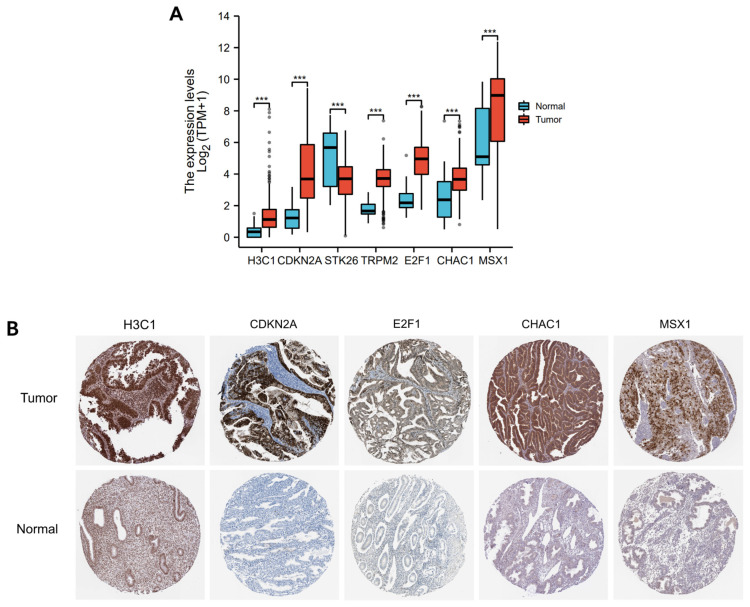
The protein level of the 7 OSRGs in UCEC tissues and normal tissues: (**A**) mRNA levels of the 7 OSRGs in TCGA; (**B**) Immunohistochemistry verification of seven OSRGs’ expression in the HPA database. Note: UCEC: uterine corpus endometrial carcinoma; OSRGs: oxidative stress-related genes. *** represents *p* < 0.0001.

**Figure 10 cancers-14-05572-f010:**
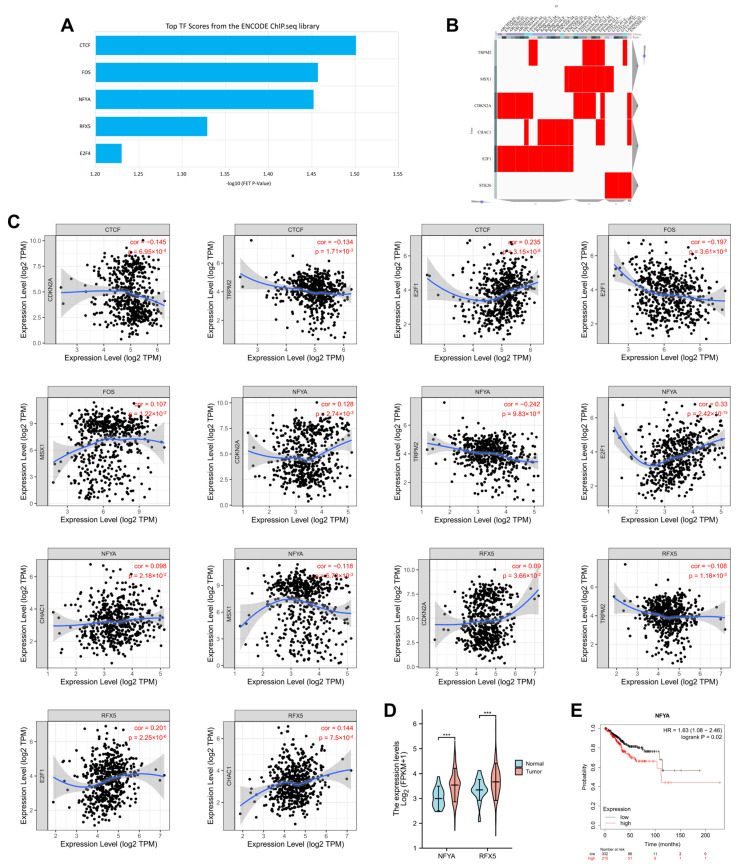
Transcription factors of 7 OSRGs in UCEC: (**A**–**C**) Top 5 transcription factors showed significant correlations with the 7 OSRGs; (**D**) mRNA levels of NFYA and RFX5 in UCEC tissues and normal tissues; (**E**) Survival analysis of NFYA in UCEC. Note: UCEC: uterine corpus endometrial carcinoma; OSRGs: oxidative stress-related genes. *** represents *p* < 0.0001.

**Table 1 cancers-14-05572-t001:** GO enrichment and KEGG analysis of 135 differentially expressed OSRGs.

Ontology	ID	Description	Gene Ratio	Bg Ratio	*p* Value	*P.* Adjust	*Q* Value
BP	GO:0031497	chromatin assembly	36/135	165/18,670	7.75 × 10^−44^	2.34 × 10^−40^	1.77 × 10^−40^
BP	GO:0097193	intrinsic apoptotic signaling pathway	42/135	289/18,670	2.09 × 10^−43^	3.15 × 10^−40^	2.39 × 10^−40^
BP	GO:0006323	DNA packaging	37/135	210/18,670	2.34 × 10^−41^	1.81 × 10^−38^	1.37 × 10^−38^
CC	GO:0000786	Nucleosome	37/135	107/19,717	1.93 × 10^−54^	4.10 × 10^−52^	3.81 × 10^−52^
CC	GO:0044815	DNA packaging complex	37/135	115/19,717	4.67 × 10^−53^	4.98 × 10^−51^	4.62 × 10^−51^
CC	GO:0032993	protein-DNA complex	38/135	202/19,717	1.53 × 10^−44^	1.08 × 10^−42^	1.01 × 10^−42^
MF	GO:0031492	nucleosomal DNA binding	8/134	55/17,697	7.90 × 10^−9^	2.63 × 10^−6^	2.30 × 10^−6^
MF	GO:0031491	nucleosome binding	8/134	85/17,697	2.58 × 10^−7^	3.32 × 10^−5^	2.89 × 10^−5^
MF	GO:0031490	chromatin DNA binding	9/134	119/17,697	2.99 × 10^−7^	3.32 × 10^−5^	2.89 × 10^−5^
KEGG	hsa05322	Systemic lupus erythematosus	40/108	136/8076	6.16 × 10^−45^	1.32 × 10^−42^	9.01 × 10^−43^
KEGG	hsa05034	Alcoholism	38/108	187/8076	8.11 × 10^−36^	8.72 × 10^−34^	5.93 × 10^−34^
KEGG	hsa05203	Viral carcinogenesis	28/108	204/8076	3.05 × 10^−21^	2.18 × 10^−19^	1.49 × 10^−19^

Note: OSRGs: oxidative stress-related genes; GO: Gene ontology; KEGG: Kyoto Encyclopedia of Genes and Genomes.

## Data Availability

All data analyzed in this paper can be found in TCGA (https://portal.gdc.cancer.gov/ accessed on 23 September 2022), HPA (https://www.proteinatlas.org/ accessed on 23 September 2022), and CHEA3 (https://maayanlab.cloud/chea3/ accessed on 23 September 2022). Patient consent was waived for the reason that it is not applicable.

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
