# Peer review of "Construction of Oxidative Stress-Related Genes Risk Model Predicts the Prognosis of Uterine Corpus Endometrial Cancer Patients"

_cancers, 2022, doi:10.3390/cancers14225572_

Round 1

Reviewer 1 Report

The authors performed multi-gene analyses to elucidate the correlation between oxidative stress and endometrial carcinogenesis. The scientific interest in this study is high, but I have some questions for the authors. 

Major comments

1

The authors first screened differential gene expression of normal and cancer tissues and set the threshold to "a log2|FC| of >1.5". However, they did not show why this value was appropriate. Is there any scientific evidence that selected genes promoted or inhibited oxidative stress when their expression increased or decreased in log2 1.5? If yes, the authors should show the references. If not, a validation study is needed to demonstrate the correlation between RNA expression, protein expression, and their effect on oxidative stress. For protein expression, they showed immunohistochemistry in 9a. I think they should show figure 9a in the former, in "3.1. Identification of oxidative stress-related DEGs in UCEC". In that case, because the protein expression of STK26 and TRPM2 was the same between normal and cancer tissues, these two genes should be excluded from the following analysis. 

2

Throughout the analysis, the authors only show statistical values to show the difference in gene expression. I request the authors to add quantitative analysis. Otherwise, readers can not understand the impact. For example, CTLA4 in normal tissue looks similar to cancer tissues in figure 7d.

Minor comment

Abbreviations are not shown in some sentences where they appeared first.

2  

Table 1 

What is bgratio?

3  

Figure 7

The authors should correct the figure number according to the main text. 

Reviewer 2 Report

This study evaluated the prognostic value of oxidative stress response genes in endometrial carcinoma (EC) using bioinformatics analysis.

I have the following comments to the Authors:

·      Line 14 and 309: please correct “accelerate” instead of “accelerates”

·      Discussion: Authors should also discuss about limitations of this study, in order to stimulate further studies about this topic.

·      Discussion: Authors wrote that “it is difficult to predict survival outcomes after conventional diagnosis and treatment. Consequently, finding reliable biological markers to predict UCEC prognosis is imperative”. In fact, molecular predictors of prognosis of EC do exist and are described also in the latest European and American guidelines for the management of EC (ESGO/ESTRO/ESP and NCCN). Authors should include in the discussion known molecular prognostic factors of EC detected by TCGA (e.g. PMID: 32377987).

Reviewer 3 Report

This article includes some interesting data about the role of oxidative stress genes in the development of endometrial carcinoma (EC) and in overall survival. In fact, genetic factors related to EC diagnosis and prognosis are a trending topic in literature and oxidative stress genes seem to have the potential to be added to the list. This study demonstrates that oxidative stress genes are worth to be further analyzed in EC.

I have the following comments to the Authors:

·      Discussion: The ESTRO/ESGO/ESP guidelines for the management of EC proposed a novel risk stratification model including molecular TCGA molecular groups  to assess the prognosis of EC and the role of these molecular subtypes of EC as prognostic factors independent from classic, well-known, clinicopathologic ones in EC (such as myometrial invasion, histotype or lymph vascular space invasion) is a very hot topic in literature to date. Authors may expand the discussion about this including the latest evidence in literature (e.g. PMID: 34088515; PMID: 34073635).

·      Discussion: In addition to molecular factors for the prognosis assessment, metabolomics has recently appeared as a promising test for a non-invasive diagnosis of several diseases and metabolites were found able to predict the presence of EC tumor behavior (progression and recurrence) and pathological characteristics (histotype, myometrial invasion and lymph vascular space invasion). Apart from this, other circulating biomarkers are described in literature for the early and non-invasive diagnosis of EC. This study demonstrates that oxidative stress response genes have the potential to be added to the list. Authors should include in the discussion a brief revision of these novel circulating predictors (e.g. PMID: 36139068), which could have an extraordinary impact on the management of EC in the future.

Round 2

Reviewer 1 Report

The authors answered almost all of my comments well. As they added a sentence to the conclusion part, the limitation of this study is the lack of quantitative analysis of the gene or protein expressions and has not been resolved in the revised version. I ask the editor to judge whether or not this limitation is disadvantage for the quality of journal cancers.